# Voltage-driven motion of nitrogen ions: a new paradigm for magneto-ionics

Julius de Rojas [1], Alberto Quintana [2], Aitor Lopeandía[1], Joaquín Salguero[3], Beatriz Muñiz[3], Fatima Ibrahim[4], Mairbek Chshiev [4,5], Aliona Nicolenco[1], Maciej O. Liedke [6], Maik Butterling[6], Andreas Wagner [6], Veronica Sireus[1], Llibertat Abad[7], Christopher J. Jensen [2], Kai Liu [2], Josep Nogués[8,9], José L. Costa-Krämer [3], Enric Menéndez [1✉] & Jordi Sort[1,9✉]

Magneto-ionics, understood as voltage-driven ion transport in magnetic materials, has largely relied on controlled migration of oxygen ions. Here, we demonstrate room-temperature voltage-driven nitrogen transport (*i.e.*, nitrogen magneto-ionics) by electrolyte-gating of a CoN film. Nitrogen magneto-ionics in CoN is compared to oxygen magneto-ionics in $Co_3O_4$. Both materials are nanocrystalline (face-centered cubic structure) and show reversible voltage-driven ON-OFF ferromagnetism. In contrast to oxygen, nitrogen transport occurs uniformly creating a plane-wave-like migration front, without assistance of diffusion channels. Remarkably, nitrogen magneto-ionics requires lower threshold voltages and exhibits enhanced rates and cyclability. This is due to the lower activation energy for ion diffusion and the lower electronegativity of nitrogen compared to oxygen. These results may open new avenues in applications such as brain-inspired computing or iontronics in general.

[1] Departament de Física, Universitat Autònoma de Barcelona, 08193 Cerdanyola del Vallès, Spain. [2] Department of Physics, Georgetown University, Washington, DC 20057, USA. [3] IMN-Instituto de Micro y Nanotecnología (CNM-CSIC), Isaac Newton 8, PTM, 28760 Tres Cantos, Madrid, Spain. [4] Univ. Grenoble Alpes, CEA, CNRS, Spintec, 38000 Grenoble, France. [5] Institut Universitaire de France, 75231 Paris, France. [6] Institute of Radiation Physics, Helmholtz-Zentrum Dresden–Rossendorf, Dresden 01328, Germany. [7] Institut de Microelectrònica de Barcelona, IMB-CNM (CSIC), Campus UAB, 08193 Bellaterra, Spain. [8] Catalan Institute of Nanoscience and Nanotechnology (ICN2), CSIC and BIST, Campus UAB, Bellaterra, 08193 Barcelona, Spain. [9] Institució Catalana de Recerca i Estudis Avançats (ICREA), Pg. Lluís Companys 23, 08010 Barcelona, Spain. ✉email: enric.menendez@uab.cat; jordi.sort@uab.cat

Magneto-ionics[1–17], i.e., the change in the magnetic properties of materials due to electric field-induced ion motion, is acquiring a leading role, among other magnetoelectric mechanisms (intrinsic[18] or extrinsic[19] multiferroicity, electric charge accumulation[20–22]), to control magnetism with voltage[23,24]. This is triggered by its capability to largely modulate magnetic properties in a permanent and energy-efficient way[2,25]. Usually, magneto-ionic systems comprise layered heterostructures built around a ferromagnetic target material, such as Co or Fe, grown adjacent to solid-state electrolyte films (e.g., $GdO_x$[2] or $HfO_2$[26]). Depending on the voltage polarity, these electrolytes accept or donate oxygen, acting as ion reservoirs. In this way, for instance, the effective magnetic easy axis of ferromagnetic layers can be precisely controlled[2]. However, room-temperature ionic response is slow ($\sim 10^2$–$10^3$ s)[2]. Therefore, since ion migration is a thermally activated process[27], ion diffusion is commonly assisted with heat[2], which is detrimental in terms of energy efficiency. Sometimes cyclability is limited due to cumulative irreversible changes that the target materials undergo from structural/compositional viewpoints[2]. Recently, through a proton-based route, $10^{-1}$ s ionic motion has been demonstrated at room-temperature, with good endurance, despite restricted hydrogen retention[9]. Hydrogen is mainly adsorbed rather than absorbed, which imposes stringent limitations on the thickness of the ferromagnet. Other approaches relying on the insertion/removal of ions, such as Li[11–13,28] or F[14], into a ferromagnet are promising in terms of reversibility. However, due to incompatibilities with complementary metal-oxide semiconductor (CMOS) architectures, applications in electronics in these cases are limited[9].

An alternative approach is to use target materials, which are already oxidized. Magneto-ionics using structural oxygen (i.e., oxygen incorporated in the crystallographic structure of the actuated material) exhibits outstanding stability and reversibility[7,8]. This has been demonstrated in electrolyte-gated[20,29–33], thick ($\geq 100$ nm) paramagnetic $Co_3O_4$ films, in which room-temperature voltage-controlled ON-OFF ferromagnetism has been achieved, benefiting from defect-assisted voltage-driven transport of structural oxygen. Nevertheless, there is an inherent voltage trade-off between induced magnetization, speed and cyclability. Specifically, the generated magnetization increases with voltage, whereas cyclability degrades for exceedingly high voltages due to irreversible losses of oxygen (i.e., bubbling)[7,8].

Here, nitrogen magneto-ionics is demonstrated as an improved alternative to oxygen magneto-ionics. CoN and $Co_3O_4$ single-layer films are voltage-actuated to compare nitrogen vs. oxygen magneto-ionic performances. These materials were selected since they both exhibit voltage-induced ON-OFF ferromagnetic transitions. Both films were grown by sputtering, have the same thickness and exhibit a similar (nanocrystalline, face-centered cubic) microstructure. Remarkably, voltage-driven transport of structural nitrogen is energetically more favorable than oxygen, resulting in lower operating voltages and enhanced cyclability. This together with the lower electronegativity (i.e., weaker bonds with Co) of nitrogen with respect to oxygen leads to overall enhanced magneto-ionic effects. Controlled motion of nitrogen ions with voltage might enable the use of magneto-ionics in new technological areas that require endurance and moderate operation speeds (e.g., neuromorphic computing[34] or micro-electro-mechanical systems[35]).

## Results

### Oxygen vs. nitrogen magneto-ionics: magnetoelectric characterization.
Structural and magnetic characterization of the as-deposited films reveals that both $Co_3O_4$ and CoN films are polycrystalline (with cubic structure) and non-ferromagnetic (Supplementary Information, Figs. S1–S4). The voltage actuation is carried out via electrolyte-gating using an electrochemical capacitor configuration[8] (Fig. 1a). In this way, the overall film area exposed to the liquid electrolyte[29,30] is activated, establishing a well-defined out-of-plane electric field[8]. To investigate magneto-ionic motion, the films were electrolyte-gated at –50 V for 12 h and, during this time, consecutive magnetic hysteresis loops (25 min duration each) were sequentially recorded. In Fig. 1b, c, the red loops correspond to the first measurement under voltage and the black arrows indicate the first sweeping leg of the cycle. In this timeframe (during the descending branch of the loop), the magnetization ($M$) of the CoN film significantly increases, whereas $Co_3O_4$ still remains paramagnetic, evidencing that nitrogen motion is significantly faster than oxygen transport. This is demonstrated in Fig. 1d, which shows the saturation magnetization, $M_S$, as a function of time (Supplementary Fig. S5 shows details on $M_S$ quantification). $M_S$ evolves with time for both films, reaching, after the magneto-ionic motion has stabilized, maximum values of 588 and 637 emu cm$^{-3}$ for $Co_3O_4$ and CoN, respectively (Table 1). By linearly fitting $M_S$ vs. $t$ during the first minutes of voltage application (wherein $M_S$ in CoN fully saturates), magneto-ionic rates of 467 and 2602 emu cm$^{-3}$ h$^{-1}$ are obtained for oxygen in $Co_3O_4$ and nitrogen in CoN, respectively (Table 1), i.e., a five-fold enhancement for the latter. For both systems, when gating at –50 V, bubbling occurs. Oxygen (in $Co_3O_4$) and nitrogen (in CoN) gas evolution is most likely the major source of bubbling. When a large negative voltage is applied, high concentrations of oxygen and nitrogen anions accumulate near the counter-electrode. If their concentration is high enough, bubbles form when the oxygen (or nitrogen) solubility limits[36,37] are exceeded. When this happens, these elements are irreversibly lost from the system (sample + electrolyte) and this is the main reason causing the lack of reversibility of magneto-ionic effects when positive voltage is applied. Partial decomposition of electrolyte gate (i.e., propylene carbonate) at high voltages cannot be completely ruled out[38], although no electron-transfer reactions can be observed in the cyclic voltammetry curve when using Pt as working electrode (Supplementary Fig. S6) and no bubbling was observed in this case. Bubbling is, in fact, more pronounced for CoN than for $Co_3O_4$, where the magneto-ionic response is stronger. The hysteresis cycles from electrolyte-gated CoN are more square-shaped than for $Co_3O_4$ (Fig. 1b, c). They exhibit larger squareness [defined as the ratio between the remnant magnetization ($M_R$) and $M_S$ ($M_R/M_S$)], and larger slopes at the coercivity ($H_C$) normalized to $M_S$ [d$M$/d$H$($H = H_C$) $M_S^{-1}$] than the loops of $Co_3O_4$ throughout the time the voltage was applied (Fig. 1e).

Additionally, the $H_C$ corresponding to $Co_3O_4$ scales monotonically with time, while in CoN $H_C$ shows a maximum at the initial stages of gating and decreases afterwards (Supplementary Fig. S7). The case of CoN bears a resemblance to the characteristic dependence of $H_C$ with particle size in magnetic systems, consistent with a homogeneous generation of ferromagnetic regions, uniformly evolving in size, starting from a superparamagnetic behavior (zero $H_C$), followed by a single domain state (maximum $H_C$) and, afterwards, reaching a multi-domain state with reduced $H_C$[39]. The squareness of the loops and the rather small $H_C$ for CoN indicate an in-plane anisotropy and likely a reversal by domain wall motion, which hints at the uniformity of the generated metallic Co phase (Table 1). Such low coercivities might be also associated with a highly nanostructured or even amorphous-like Co[40]. In contrast, the generated ferromagnetism in $Co_3O_4$ shows much larger $H_C$, which results from the formation of isolated Co clusters immersed in a residual $Co_3O_4$ matrix[7,8].

The onset voltage for magneto-ionic motion was determined in as-prepared films by monotonically increasing the absolute value

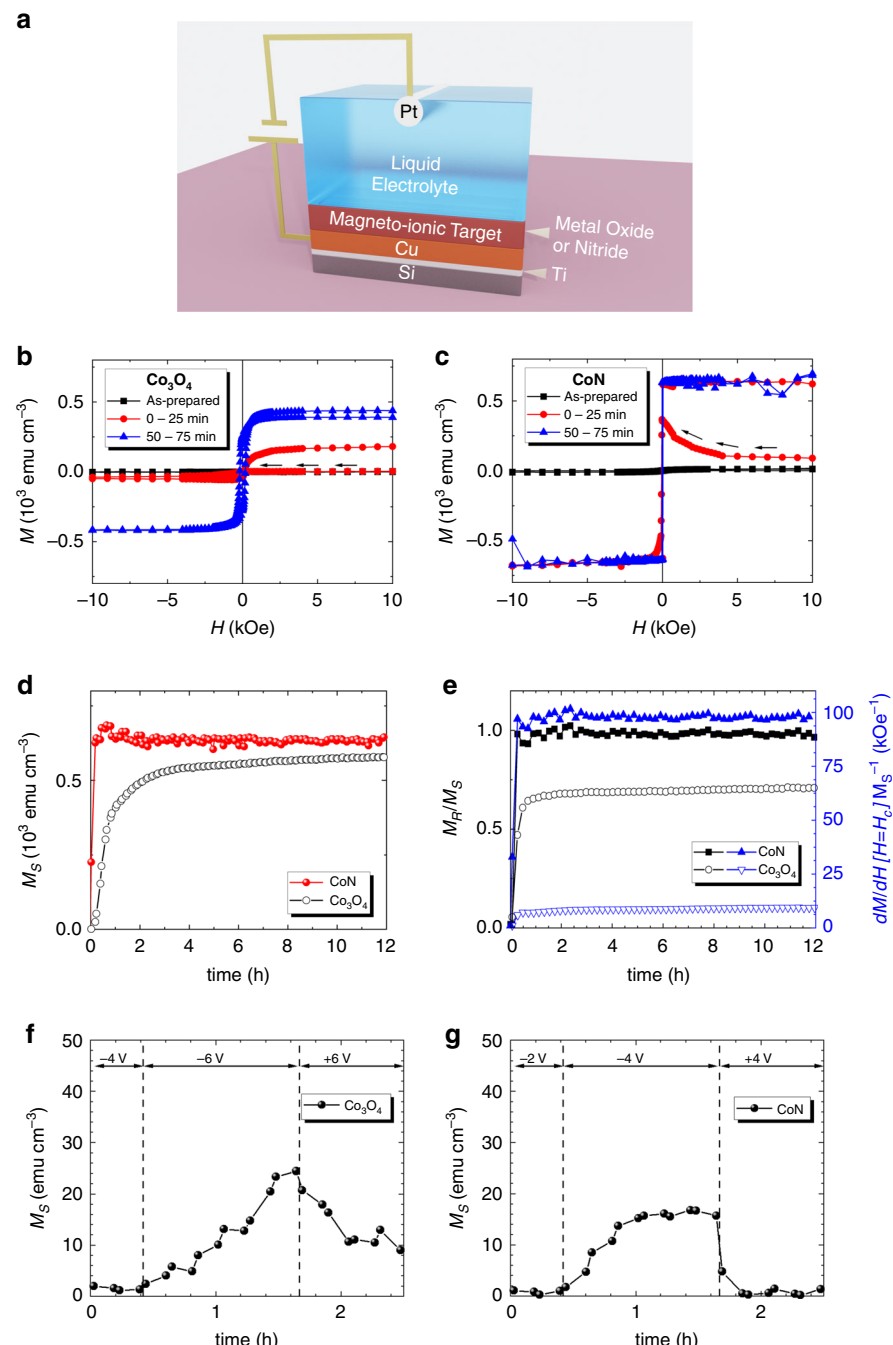

**Fig. 1 Oxygen vs. nitrogen magneto-ionics: magnetoelectric characterization. a** Schematic representation of the electrolyte-gating process in the $Co_3O_4$ and CoN films. **b, c** Hysteresis loops (each of 25 min of duration) of the as-prepared film (black) and under –50 V gating (the first (0–25 min) and the third (50–75 min) cycles are shown only) for the $Co_3O_4$ and CoN films, respectively, obtained by in-plane vibrating sample magnetometry. **d** Time evolution of the saturation magnetization ($M_S$ vs. $t$) and **e** squareness ($M_R/M_S$) and slope of hysteresis loop at $H_C$ normalized to $M_S$ ($dM/dH[H = H_C] M_S^{-1}$) for each film. **f, g** Time evolution of $M_S$ when the gating is monotonically increased in steps of –2 V to determine the onset voltage required to display ferromagnetism for $Co_3O_4$ and CoN, respectively.

**Table 1 Oxygen vs. nitrogen magneto-ionics by magnetoelectric measurements.**

| Film | Onset voltage (V) | Recovery voltage (V) | dM/dt @ –50 V (emu cm$^{-3}$ h$^{-1}$) | $M_S$ (emu cm$^{-3}$) | $M_R/M_S$ (%) | dM/dH $M_S^{-1}$ @ $H_C$ (kOe$^{-1}$) | $H_C$ (Oe) |
|------|-------------------|----------------------|------------------------|----------------|---------------|--------------------------------|------------|
| $Co_3O_4$ | –6 | +20 | 467 | 588 | 62 | 7.4 | 185 |
| CoN | –4 | +4 | 2602 | 637 | 96 | 97.4 | 17 |

Onset and recovery voltages, magneto-ionic speed and magnetic properties of the generated ferromagnetic phases. Note that $M_S$ has been obtained by normalizing the magnetic moment to the nominal film thickness, without taking into consideration the formation of vacancies or eventual nanoporosity.

of the negative gating in steps of –2 V to observe when the films started to display ferromagnetism (Fig. 1f, g and Supplementary Fig. S3). Interestingly, the onset voltage for CoN (–4 V) is lower than for $Co_3O_4$ (–6 V). To investigate reversibility of the magneto-ionic process, both systems were kept at their respective onset voltages for one hour and, then, the polarity was inverted (i.e., +4 and +6 V). Remarkably, while $Co_3O_4$ recovers only partially at +6 V, CoN fully recuperates the pristine paramagnetic state at +4 V. These results anticipate a higher activation energy for oxygen transport than for nitrogen. In fact, to recover the paramagnetic state in $Co_3O_4$ + 20 V are required (Table 1). The need to actuate at a higher voltage induces irreversible losses of oxygen since the liquid electrolyte has a limited solubility of oxygen[41]. Note that if –50 V are applied, none of the treated films is recoverable in agreement with the irreversible loss of oxygen and nitrogen through bubbling.

**Magneto-ionic cyclability.** To investigate the cyclability, the CoN and $Co_3O_4$ films were subjected to –4 V/ +4 V and –8 V/ +8 V pulses of relatively short duration (≈8.5 min/cycle). The duration of the onset negative voltage pulse was selected in each material to give a $\Delta M$ of approximately 1 emu cm$^{-3}$ in the first cycle. This resulted in times of approximately 4.1 min for CoN and 6.2 min for $Co_3O_4$ (evidencing that the response of $Co_3O_4$ is delayed with respect to that of CoN). As seen in Fig. 2, cycling at –/+4 V results in a very stable periodic response for CoN. Conversely, at this voltage, no traces of magneto-ionic effects are observed in $Co_3O_4$, corroborating the need of a higher onset voltage. Good cyclability is observed for $Co_3O_4$ at –8 V/ +8 V. Note that, in contrast to Fig. 1f, short pulses at –/+8 V allow recovering the initial state in $Co_3O_4$ after each cycle. Both materials show no progressive irreversible gain/loss of magnetic signal upon successive cycling. However, while the $\Delta M$ amplitude in each cycle remains stable for CoN, a loss of about 25% is observed in $Co_3O_4$ after the first cycle. As a first approach, a simple calculation considering the overall magnetization induced in CoN when applying – 50 V (i.e., 637 emu cm$^{-3}$, Table 1, when the whole CoN is affected, as will be shown in the forthcoming section) suggests that during this reversible cycling (1 emu cm$^{-3}$, Fig. 2), only the uppermost 0.1–0.2 nm of the CoN layer are involved in the observed reversible effect. If the threshold negative voltage is applied for longer times (Fig. 1g), the induced magnetization from nitrogen ion migration leads to an equivalent CoN

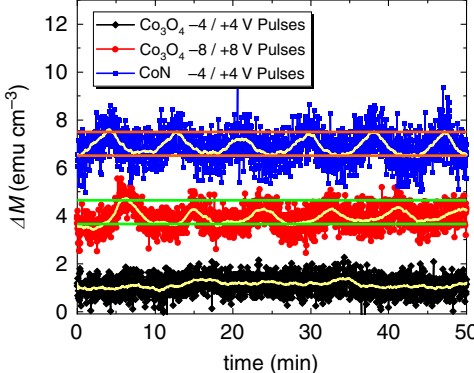

**Fig. 2 Cyclability.** Magneto-ionic cyclability of the $Co_3O_4$ and CoN films subjected to -4 V/+4 V and -8 V/+8 V, and to -4 V/+4 V, respectively, with pulses of short duration (≈8.5 min/cycle). The pale yellow lines represent the average curves. Horizontal lines (red, green) are guides to the eye and span 1 emu cm$^{-3}$. The data are shifted in $\Delta M$-axis to make them distinguishable. Cyclability was carried out under the application of 5 kOe to ensure being above the anisotropy field and, thus, in saturation.

thickness of 2–3 nm. Similar affected thicknesses during reversible voltage-driven magnetization cycling can be estimated for $Co_3O_4$. In any case, overall, Fig. 2 corroborates that magneto-ionic rates, onset voltages and endurance are better when using nitrogen than oxygen migration.

**Ion transport mechanisms.** Cross-section lamellae of the as-prepared and electrolyte-gated samples treated at –50 V for 75 min were studied by transmission electron microscopy (TEM) (Fig. 3). The as-prepared $Co_3O_4$ shows regular, columnar-shaped grains (Fig. 3a and Supplementary Fig. S8a). Co (red) and O (blue) are homogeneously distributed in the as-grown film (Fig. 3b). Conversely, upon gating at –50 V, this morphology drastically changes redistributing the elements into O-rich channels in agreement with previously reported results[7,8] (Fig. 3c, d and Supplementary Fig. S7c). This confirms that oxygen transport takes place via a two-fold mechanism: (i) uniform oxygen transport towards the electrolyte and (ii) localized oxygen migration along diffusion channels. High-resolution TEM (Fig. S9) shows remaining $Co_3O_4$ after applying –50 V. Electric-field induced oxygen ions exchange with the liquid electrolyte is the main reason of the increased current densities observed in the cycling voltammetry curves (Supplementary Fig. S10).

The as-prepared CoN film shows an isotropic and highly nanostructured morphology with homogeneous composition (Fig. 3e, f and Supplementary Fig. S7b). Remarkably, upon gating at –50 V for 75 min, a well-defined interface (resembling a diffusion front) parallel to the surface is distinguished, dividing the film in two sublayers with different microstructures (Supplementary Figs. S8d and S11). No traces of nitrogen are detected in either of the two sublayers, evidencing a full denitriding process, consistent with the magnetoelectric characterization. While sublayer 2 is nanocrystalline, sublayer 1 (in contact with the electrolyte during magnetoelectric measurements) is amorphous-like and exhibits lower density (decrease of Co signal in Fig. 3h), possibly due to free volume or nanoporosity. This suggests a complex denitriding process in which, under large negative voltages, not only does nitrogen migrate from the sample to the electrolyte, but also Co is redistributed across the film thickness. Similar to $Co_3O_4$, these processes are essentially non-Faradaic and do not cause pronounced peaks in the cyclic voltammetry curves (Supplementary Fig. S10).

To further asses the microstructure of the films upon magneto-ionic actuation, variable energy positron annihilation lifetime spectroscopy experiments[42,43] were performed (Methods). In the as-prepared CoN sample, only $\tau_1$ and $\tau_2$ lifetime components are observed, indicating the absence of void-like structures (no $\tau_3$) in the pre-biasing state (Fig. 4). $\tau_1$ is around 0.28 ns in the top half of the film, which could correspond to a cluster of more than 4 vacancies[7]. For the bottom half, $\tau_1$ is slightly lower, indicating clusters with less than 4 vacancies. The second lifetime $\tau_2$ represents a mixture of signals from surface states and grain boundaries. In the first tens of nm, $\tau_2$ is larger than 0.5 ns, indicating the presence of small voids. As seen in Fig. 4b, the relative intensity $I_1$ decreases, while $I_2$ increases reaching similar intensities at the interface with the buffer layer, reflecting the influence of the buffer polycrystallinity in the CoN growth, whose extent decreases with film thickness.

Negative biasing of –20 V and –50 V increases $\tau_1$ and $\tau_2$ and this increase scales with voltage, indicating that the initial open volumes become larger. Already for –20 V, $\tau_3$ emerges, showing the presence of large voids as it happens in $Co_3O_4$[7]. However, for $Co_3O_4$, a monotonic increase of relative intensity $I_2$ across the film thickness is observed[7], whereas relative maxima (marked with arrows in Fig. 4b) are found for CoN. The depth position of

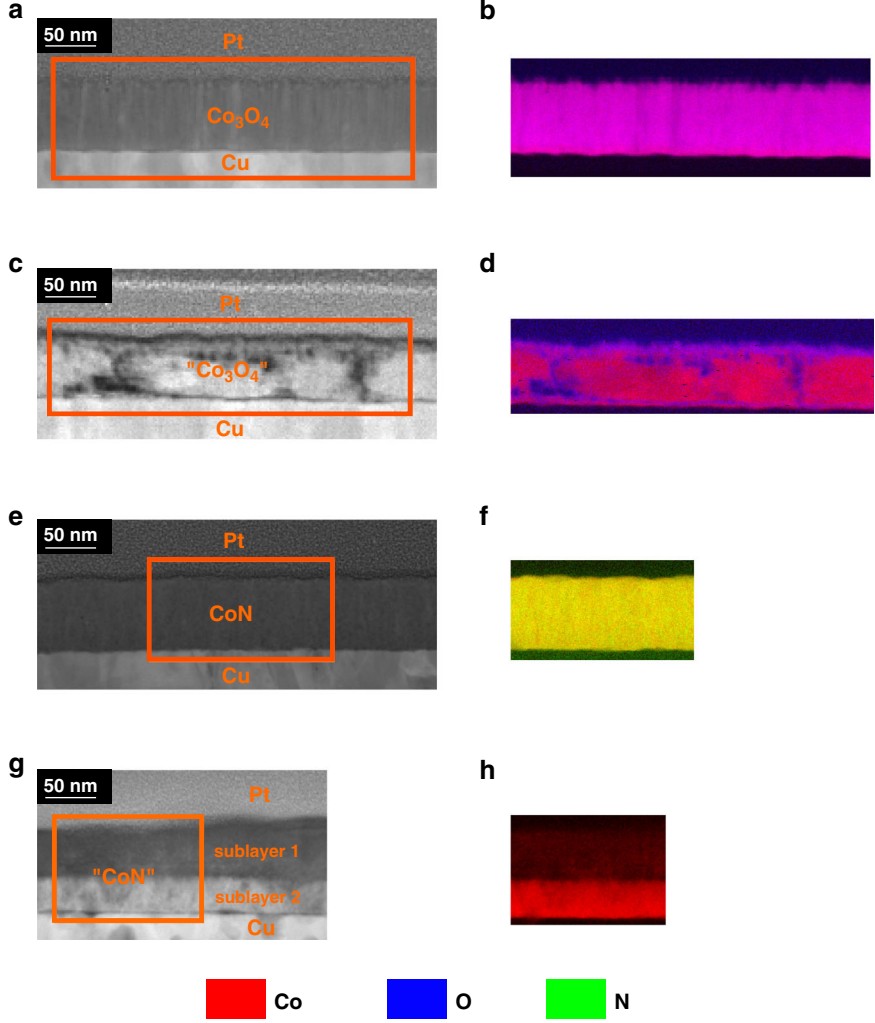

**Fig. 3 Compositional characterization by high-angle annular dark-field scanning transmission electron microscopy (HAADF-STEM) and electron energy loss spectroscopy (EELS). a–b**, **c–d**, **e–f**, and **g–h** are the HAADF-STEM images and corresponding elemental EELS mappings of the areas marked in orange, respectively, of the as-prepared $Co_3O_4$ film, $Co_3O_4$ film subjected to a −50 V for 75 min, as-prepared CoN film and CoN film subjected to a −50 V for 75 min, respectively. The colors corresponding to each element for the EELS analyses are depicted at the bottom of the figure.

these relative maxima increases with the applied voltage, suggesting the occurrence of an interface migration front, in agreement with TEM observations. For −50 V, the migration front moves deeper into the film in agreement with a more intense denitriding/amorphization process. $I_3$ tends to decrease and vanishes with thickness, evidencing that larger voids are only present at the top surface (as hinted by the TEM-EELS analysis; Fig. 3h). The ionic transport upon electrolyte-gating in CoN is thus consistent with a uniform nitrogen migration through vacancies and grain boundaries ($\tau_1$ and $\tau_2$ increase), leaving behind larger grain boundaries and voids.

**Co-O vs. Co-N formation energy.** Neither $Co_3O_4$ nor CoN are ferromagnetic at room-temperature. The magnetic properties of these systems are strongly correlated to the amount of either oxygen or nitrogen in the films. According to the virtual crystal approximation, for CoN, beyond 50 at. % of nitrogen in the unit cell, the magnetic moment becomes negligible (Fig. 5a). This explains why the as-prepared CoN film is not magnetic and it also sets a limit for irreversible losses of nitrogen beyond which successive voltage-driven ON-OFF-ON ferromagnetism would be compromised.

To simulate the Co-O and Co-N formation energy (which occurs when the dissolved oxygen and nitrogen ions are

re-introduced to the films with positive voltage), the insertion of an atom of either oxygen or nitrogen into a cobalt slab, with either (0001) HCP or (111) FCC orientation, has been considered. Minimum energy paths were calculated by the nudged elastic band method (Methods). The obtained total energies per atom normalized to the global minimum value are plotted in Fig. 5b as a function of the displacement $z$ of the oxygen or nitrogen atom from the cobalt reference layer. The reference $z = 0$ Å is assigned to the position of the outermost cobalt layer. The global energy minimum is found at $z = 1.14$ Å for both oxygen and nitrogen cases. Another local minimum is located around $z = -0.75$ Å for oxygen and $z = -1$ Å for nitrogen. In turn, the calculated energy barriers between the two minima are 1.54 (1.85) and 1.14 (1.37) eV/atom for oxygen and nitrogen displacement into HCP (FCC) Co surface, respectively. Thus, inserting nitrogen into cobalt is energetically more favorable (i.e., less energy required) compared to inserting oxygen. Using the calculated energy barrier values, one can infer the critical electric fields, $E_C$, needed to overcome the energy barrier by oxygen and nitrogen atoms. These electric fields can be estimated as $E_C = \frac{\Delta V}{\Delta z}$ where $\Delta V$ and $\Delta z$ represent, respectively, the electric potential per atom to overcome the energy barrier and the distance between minima that an atom must migrate[25]. $E_C$ is found to be 8.1 (8.5) V nm⁻¹ and

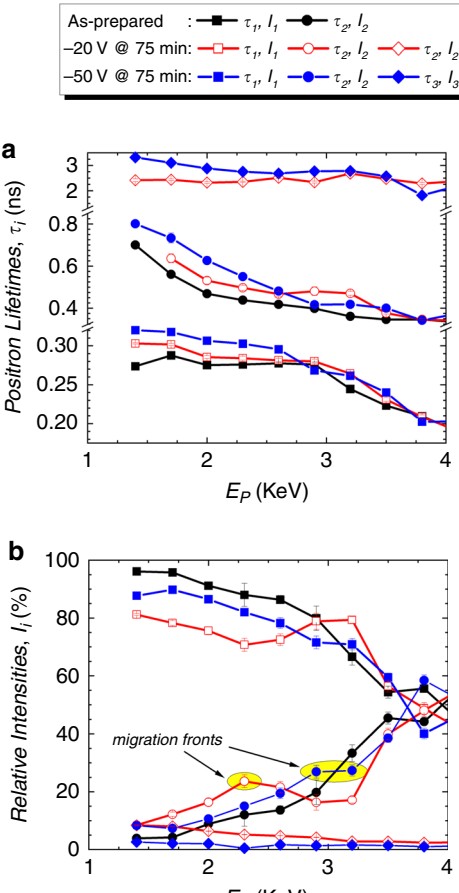

**a**

**b**

**Fig. 4 Defect characterization by variable energy positron annihilation lifetime spectroscopy (VEPALS). a** Positron lifetime components $\tau_{i=1-3}$ and **b** their relative intensities $I_{i=1-3}$ as a function of positron implantation energy $E_p$ for as-prepared and –20 V and –50 V biased CoN films. The non-monotonic change of intensity $I_2$ with depth is linked to the position of the interface between Co sublayers reminiscent of an ionic migration front as highlighted in **b**.

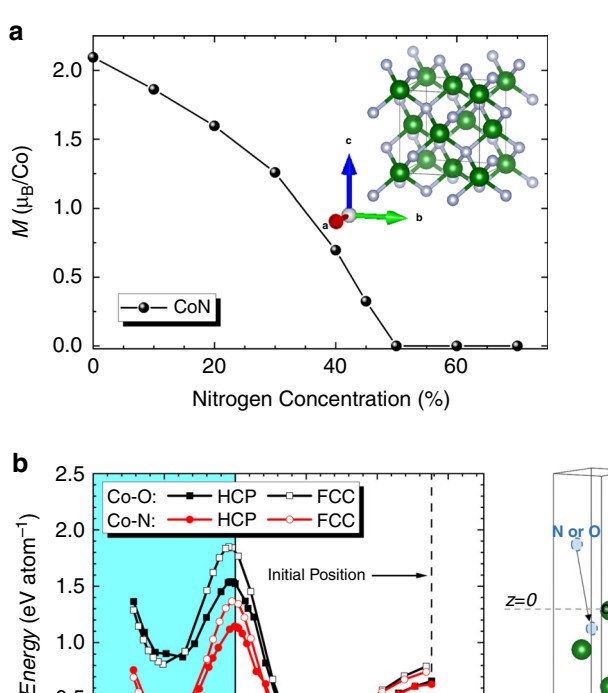

**a**

**b**

**Fig. 5 Ab initio calculations: magnetism in the Co-N system and Co-O vs. Co-N formation energy. a** Variation of magnetic moment in Bohr magneton, $\mu_B$, per Co atom as a function of increasing the N percentage in the CoN unit cell, shown in the inset, calculated within Virtual Crystal Approximation (VCA). **b** Calculated total energy per atom, normalized to the minimum energy value, as a function of the displacement between the reference Co outermost surface atom and the inserted O or N atom. Both HCP (0001) and FCC (111) surfaces are considered represented by filled and open symbols, respectively. The black squares (red circles) correspond to the O (N) energetic path, respectively. The five-monolayer-thick Co slab is shown in the right panel, where the dashed line indicates the reference $z$ position, which is the outermost Co surface monolayer.

5.3 (6.3) V nm⁻¹ for oxygen and nitrogen migration into HCP (FCC) Co, respectively, in good agreement with the onset voltages from magnetoelectric measurements (Fig. 1f, g) (thickness of the electric double layer <1 nm). It is important to point out that due to the limitations of the applied density-functional theory and in particular the nudged elastic band method, the CoN and Co₃O₄ crystallographic structures were not reproduced, starting initially from the Co film while calculating the presented energy considerations. However, with the chosen method we were able to provide both a good explanation of the ionic energetics and reasonable agreement with the experiment. More importantly, the calculations allowed estimating the critical electric field needed to overcome the barrier. It is worth noting that the same method was efficiently used to describe the voltage control of the magnetic anisotropy by O migration at Fe/MgO[25]. Therefore, we believe that the applied approach is plausible to compare O vs. N magneto-ionics.

Since the ionic radii of nitrogen ions are larger than the ionic radius of $O^{2-}$, once the energy barrier for ion diffusion is overcome ionic motion would be, in first approximation, expected to be larger for oxygen than for nitrogen. However, the simulation results indicate the opposite, revealing that other parameters, such as electronegativity, might play a more dominant role than ion size in ionic motion. In fact, the Pauling

electronegativity of nitrogen is lower than that of oxygen, resulting in weaker bonds with cobalt, allowing for an enhanced ionic motion.

## Discussion

Our work demonstrates robust room-temperature nitrogen magneto-ionics in CoN. Nitrogen magneto-ionics shows reduced activation energies for ionic transport, thus requiring lower voltage actuation. Moreover, the magneto-ionic rates are faster than for oxygen magneto-ionics. This is linked to the conjugation of a lower critical electric field to overcome the energy barrier for ion diffusion and a lower electronegativity of nitrogen with respect to oxygen. The dissimilar electric properties of CoN and Co₃O₄ are also likely to play a role in the way ions diffuse in the two layers. Thus, nitrogen magneto-ionics represents a robust alternative for efficient voltage-driven effects and may enable the use of magneto-ionics in devices that require endurance and moderate speeds of operation, such as brain-inspired/stochastic computing or magnetic micro-electro-mechanical systems. The reported effects are also appealing to extend the use of nitride semiconductors in diverse applications such as electrochemical sensors, catalysis, batteries, spintronics, or iontronics.

## Methods

**Sample preparation.** Eighty-five nanometer thick $Co_3O_4$ and CoN films were grown by reactive sputtering on B-doped, highly conducting [100]-oriented Si wafers (0.5 mm thick), previously coated with 20 nm of Ti and 60 nm of Cu. Depositions were carried always out while partly masking the Cu layer to serve as working electrode.

The $Co_3O_4$ films were grown at room-temperature in an AJA International ATC 2400 Sputtering System with a base pressure in the $10^{-8}$ Torr range. High purity Ar and $O_2$ gases were used. The target to substrate distance was around 8 cm and the sputtering rate of about 5 Å s$^{-1}$. $Co_3O_4$ was grown in a 7% $O_2$/93% Ar atmosphere at a total pressure of $2.5 \times 10^{-3}$ Torr.

The CoN films were grown in a homemade triode sputtering system with a base pressure in the $10^{-8}$ Torr range. Ultra-high vacuum was ensured to minimize oxygen contamination and, thus, to rule out traces of oxygen magneto-ionics. The target to substrate distance was around 10 cm and the sputtering rate about 1 Å s$^{-1}$. CoN was grown in a 50% $N_2$/50% Ar atmosphere at a total pressure of $8 \times 10^{-3}$ Torr.

**Magnetoelectric characterization.** Magnetoelectric measurements were carried out by performing vibrating sample magnetometry while electrolyte gating the films in a capacitor configuration at room-temperature. A magnetometer from Micro Sense (LOT-Quantum Design), with a maximum applied magnetic field of 2 T, was used. The samples are mounted in a homemade electrolytic cell filled with anhydrous propylene carbonate with $Na^+$ solvated species (5–25 ppm). The magnetic properties were measured along the film plane upon applying different voltages, using an external Agilent B2902A power supply, between the Cu working electrode and the counter-electrode (Pt wire). The sign of voltage was such that negative charges accumulate at the working electrode when negative voltage was applied (and vice versa for positive voltages). The $Na^+$ solvated species in the electrolyte are aimed at reacting with any traces of water[30]. The magnetic signal is normalized to the volume of the sample exposed to the electrolyte. Note that the hysteresis loops were background-corrected using the signal at high fields (i.e., fields always far above saturation fields) to eliminate linear contributions.

**Structural and compositional measurements.** $\theta/2\theta$ X-ray diffraction (XRD) patterns were recorded on a Materials Research Diffractometer (MRD) from Malvern PANalytical company, equipped with a PIXcel1D detector, using Cu $K_\alpha$ radiation. The patterns were analyzed using a full-pattern Rietveld refinement method.

High-resolution transmission electron microscopy (HRTEM), high-angle annular dark-field scanning transmission electron microscopy (HAADF-STEM), and electron energy loss spectroscopy (EELS) were performed on a TECNAI F20 HRTEM/STEM microscope operated at 200 kV. Cross sectional lamellae were prepared by focused ion beam and placed onto a Cu transmission electron microscopy grid.

**Transport measurements.** To determine electric properties, both films ($Co_3O_4$ and CoN) were deposited onto high resistivity Si substrates. To assess the semi-conducting/metallic behavior of CoN, resistivity values were acquired from 30 to 300 K. In all cases, the van der Pauw configuration was used.

**Variable energy positron annihilation lifetime spectroscopy.** Variable energy positron annihilation lifetime spectroscopy (VEPALS) measurements were conducted at the mono-energetic positron source (MePS) beamline, which is an end station of the radiation source ELBE (Electron Linac for beams with high Brilliance and low Emittance) at Helmholtz-Zentrum Dresden-Rossendorf (Germany)[42] using a digital lifetime CeBr$_3$ scintillator detector with a homemade software employing a SPDevices ADQ14DC-2X with 14 bit vertical resolution and 2GS s$^{-1}$ (GigaSamples per second) horizontal resolution and with a time resolution function down to about 0.205 ns. The resolution function required for spectrum analysis uses two Gaussian functions with distinct intensities depending on the positron implantation energy, $E_p$, and appropriate relative shifts. All spectra contained at least $1 \times 10^7$ counts. The spectra were deconvoluted using the non-linearly least-squared-based package PALSfit fitting software[43] into discrete lifetime components, which directly confirm different defect types—i.e., sizes—(Fig. 4).

The corresponding relative intensities ($I_i$) reflect to a large extent the concentration of each defect type. In general, positron lifetime ($\tau_i$) is directly proportional to defect size, i.e., the larger the open volume, the lower the probability and the longer it takes for positrons to be annihilated with electrons[42–47]. The positron lifetime and its intensity are probed as a function of positron implantation energy $E_p$ or, in other words, implantation depth (thickness). The shortest lifetime component ($\tau_1 < 0.32$ ns) represents positron annihilation inside vacancy clusters (likely within grains). The intermediate lifetime ($0.35 < \tau_2 < 0.90$ ns) accounts for annihilation at larger vacancy clusters (linked to grain boundaries), surface states, and small voids/pores ($0.28 - 0.37$ nm in diameter, calculated based on the shape-free model for pore-size estimation of Wada et al.[48]). The longest lifetime component ($2.3 < \tau_3 < 3.3$ ns) indicates contributions of larger voids ($0.58 - 0.74$ nm in diameter).

**Ab initio calculations.** The first-principles calculations were based on the projector-augmented wave (PAW) method[49] as implemented in the VASP package[50–52] using the generalized gradient approximation[53]. We used cubic unit cell with a F43m space

group for CoN. The full structural relaxation was performed until the forces became smaller than 1 meV Å$^{-1}$, yielding a lattice constant of 4.41 Å. The virtual crystal approximation[54] was used to model the variation of nitrogen per unit cell. To compare the Co-O and Co-N formation energy, the nudged elastic band method (NEB)[55,56] was employed on the oxygen and nitrogen pathway into a five-monolayer thick (0001) hexagonal close-packed Co slab. At each step, the atomic coordinates were relaxed until the forces became smaller than 1 meV Å$^{-1}$. A kinetic energy cut-off of 500 eV was used for the plane-wave basis set and $25 \times 25 \times 25$ and $25 \times 25 \times 1$ k-point meshes were used to construct the first Brillouin zone in CoN unit cell and the Co slab in the NEB calculations, respectively.

**Reporting summary.** Further information on research design is available in the Nature Research Reporting Summary linked to this article.

## Data availability

The data used in this article are available from the corresponding authors upon request.

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

## Acknowledgements

Financial support by the European Research Council (2014-Consolidator Grant Agreement No. 648454 and 2019-Proof of Concept Grant Agreement N° 875018), the Spanish Government (MAT2017-86357-C3-1-R), the Generalitat de Catalunya (2017-SGR-292 and 2018-LLAV-00032), the French ANR (ANR-18-CE24-0017 Project "FEOrgSpin") and FEDER (MAT2017-86357-C3-1-R and 2018-LLAV-00032) is acknowledged. ICN2 is funded by the CERCA program/Generalitat de Catalunya and supported by the Severo Ochoa Centres of Excellence program (Spanish Research Agency, grant no. SEV-2017-0706). Work at GU has been supported by SMART (2018-NE-2861), one of seven centers of nCORE, a Semiconductor Research Corporation program sponsored by NIST, and NSF (DMR−1905468, DMR−1828420). The PALS measurements were carried out at ELBE at the Helmholtz-Zentrum Dresden-Rossendorf e. V., a member of the Helmholtz Association. We would like to thank A.G. Attallah and E. Hirschmann for assistance during PALS.

## Author contributions

E.M. and J.S. had the original idea and led the investigation. J.d.R., E.M. and J.S. designed the experiments. A.Q., C.J.J., and K.L. synthesized the $Co_3O_4$ films and J.S., B.M., J.N., and J.L.C. prepared the CoN films. E.M., A.Q., and K.L. designed the sample holder to carry out the magnetoelectric measurements. J.d.R. and E.M. carried out the magnetoelectric measurements and analyzed the data. J.d.R., A.Q., C.J.J., and E.M. performed the XRD characterization and analyzed the data. A.L., L.A., and E.M. carried out the TEM and STEM characterization and analyzed the corresponding data. A.N. performed the electrochemical studies. F.I and M.C. performed the ab initio calculations. J.S., B.M., J.N., and J.L.C. carried out the transport measurements. M.O.L., M.B., and A.W. characterized the samples by PALS and analyzed the data. All authors discussed the results and commented on the article. The article was written by J.d.R., E.M., and J.S.

## Competing interests

The authors declare no competing interests.
