## [Peer Review File · Nature Communications]

Reviewers' Comments:

Reviewer #1:

Remarks to the Author:

In the manuscript "Voltage-driven motion of nitrogen ions: a new paradigm for magneto-ionics" the authors set out to demonstrate a new type of a magneto-ionic (magnetoelectric) system driven by Nitrogen ions rather than conventionally used (most studied) Oxygen. The authors argue and ultimately prove that voltage-controlled nitrogen mediated redox reactions - bringing about the magnetoelectric coupling - require lower threshold voltages and exhibit enhanced rates and cyclability.

To elucidate the phenomena behind the magnetization switching and phase transitions the authors have chosen a wide-ranging and well-thought set of characterization techniques. They inquired in detail into the structure, microstructure, elemental mapping, magnetic characteristics versus applied voltage etc. The conclusions reached are sound and sufficiently supported by the experimental data. The manuscript is clearly structured with the appropriate choice of references. In particular, I find very appealing the combination of the semiconductor (or weakly metallic) material, undoubtedly resulting in the smooth and uniform redox process during the cycling, with the smart choice of the active ion.

In summary, this is a very interesting study; it should be published after addressing some remaining concerns pointed out below.

Some specific points recommended for considering:

1) As the authors concluded the full chemical reduction of the CoN film shown in the report (i.e. conversion from CoN to Co metal; for example at -50 V) is irreversible; on the other hand reversibility is demonstrated for the small voltages and shorter charging times (Fig. 2). I think it would be beneficial to discuss shortly - or in other words - an estimate of the actual thickness of the CoN participating actively in the reversible cycling. For example, let`s make a very simple (may be too simple but worth trying anyway) approximation and take the experimental change of magnetization of 1emu/cm³ (as in Fig 2.) and compare it with the total Co(N) magnetization of 637 emu/cm³ after full reduction (Table 1). Now taking 85 nm of the CoN film thickness one can estimate roughly the thickness of the magnetically active layer - $(1/637) * 85\text{nm} = 0.13\text{nm}$. Does that mean that in the reversible cycling only a few top monolayers of the CoN are involved? How deep one can actually go with the CoN reduction to retain a full reversibility? Perhaps it would be instructive for the reader to shortly address those points since at the end the strength of the magnetic response to the electric stimulus (or strength of the magnetoelectric coupling) depends on those conditions.

2) In Fig. 1f,g the reaction threshold voltages are estimated and applied for 1 hour (chemical reduction). Could you please comment on the fact that in one hour Co₃O₄ reaches the magnetization of 200 emu/cm³ while CoN shows only 15 emu/cm³. How this could be reconciled with the generally faster magnetic response of the CoN over Co₃O₄ (Fig. 1d).

3) A question out of curiosity: When all the CoN film is fully converted to Co metal, why is the measured Co magnetization after -50 V treatment (Table 1 Ms of 637) - less than half of the Co bulk value (bulk Ms of about 1400 emu/cm³)? Perhaps the authors could shortly comment on it in the manuscript.

4) Why actually CoN is claimed to be semiconducting here (page 26) ? "Fig. S4) reveal their insulating and semiconductor character, respectively. This allows the layers to hold electric fields across them, which is a necessary condition for magneto-ionic phenomena." Fig. S4 actually suggest a metallic like electronic transport down to 100 K. The CoN resistivity at room temperature is low - $4 * 10^{-4}$ Ohm cm (for instance metallic alloys nichrome is of 10^{-4} ohm cm). If the CoN is metallic-like at room temperature that would mean a creation of the Helmholtz double layer at the CoN interface (no electric field penetration into the bulk of the CoN). Perhaps this would explain why - contrary to Co₃O₄ case - the redox reaction front moves into the material in a uniform way; this would mean a conversion reaction (CoN to Co) front moving with the Helmholtz double layer into the material interior; please address this issue and maybe consider discussing it in the manuscript.

5)

Reviewer #2:

Remarks to the Author:

Remarks to the Author:

This paper reports electrical control of magnetism by voltage-driven ion transport (magneto-ionic control) for two-types of cobalt compounds; CoN and Co_3O_4 . The authors control transport of nitrogen ion or oxygen ion by electrolyte-gating method in a CoN- and Co_3O_4 -films, respectively. They have found reversible voltage-driven magnetic switching behavior between paramagnetic and ferromagnetic states at room temperature. By comparing the responsiveness of magnetization to gating voltage for CoN with that of Co_3O_4 , they have also found lower operating voltage and the enhanced cyclability in CoN. The superior electrical magnetism-controllability of CoN is clearly demonstrated, and the results would be promising for the future application of magneto-ionics in devices. However, there are several comments and questionable points, which are shown below, on this paper in its present form. In particular, the electrochemical assessment of the electrolytic cell used for the measurements seems to be insufficient. In my opinion, they should be made clear before the manuscript is suitable for publication.

(1) Redox potential measurements of the electrolytic cell should be conducted.

In this work, the authors have applied a voltage as high as -50V to the cell. Such high voltage should induce a variety of electrochemical reaction not only in magneto-ionic target (CoN and Co_3O_4 film) but also in the components of the used electrolytic cell; redox reaction of liquid electrolyte and Cu/Ti-metal film. At least, the redox potentials of CoN and Co_3O_4 should be checked by measuring cyclic voltammetry. They should directly correlate with the responsiveness of magnetization to the applied voltages.

(2) Does the Cu-layer not react with migrated oxygen or nitrogen ions?

The working electrode of electrolytic cell consists of Cu-metal. I am afraid that the Cu-layer might be oxidized under the high voltage and the oxidized Cu might react with the migrated oxygen or nitrogen ions. In such case, the formation of copper oxides and nitrides would affect the responsiveness of magnetization. Did the authors confirm the stability of the copper layer during voltage sweeping process? In case that oxidation of Cu occurs, the oxidation current would be detected during voltage sweeping process.

(3) How did the authors identify the gases produced under the gating voltage of -50V as O_2 and N_2 ?

In Page 5, the occurrence of bubbling under the gating voltage of -50V is reported for both Co_3O_4 and CoN. The authors identify the generating gases as O_2 and N_2 from Co_3O_4 and CoN, respectively. However, in such voltage as high as -50V , solvent of electrolyte (propylene carbonate) might be electrochemically decomposed to generate gases. Was the electrochemical window of electrolyte confirmed? The author should state how they distinguish the evolution of O_2/N_2 gas from that of other possible origins.

(4) Are the Na^+ -ions in liquid electrolyte inserted into the CoN/ Co_3O_4 film or not during voltage sweeping process?

The liquid electrolyte used in this work contains Na^+ -ions. In the recent years, the Na^+ -ion is intensively investigated as charge carrier species in ion-batteries. In the present case, can the authors exclude the possibility Na^+ -ion insertion into CoN/ Co_3O_4 ? In the case that the Na^+ -ion insertion occurs, Na^+ -ion insertion would contribute to redox reaction in CoN and Co_3O_4 , and thus it would affect the responsiveness of magnetization

to the applied voltages.

(5) In the Ab initio calculations, were the crystal structures of Co_3O_4 and CoN reproduced in the lattice relaxation process after oxygen or nitrogen insertion? For comparison of the formation energy of Co-O and Co-N, the calculation for the oxygen and nitrogen insertion to the hexagonal closed-packed Co slab has been conducted (In Page 19 (Methods, Ab initio calculations)). However, the sublattice-structures of Co in Co_3O_4 and CoN are different from the hexagonal closed-packed structure. In the relaxation process of the atomic coordinates, were the crystal structures of Co_3O_4 and CoN reproduced? If not, is it plausible to use the calculated values of energy barriers for explanation of the observed superior electrical magnetism-controllability of CoN?

(6) The definition of the sign of the applied voltage should be clearly stated. In page 17 (Methods, magnetoelectric characterization), it is stated that the voltage was applied in the same fashion to that presented in references 7, 8 and 27. However, the direction of the battery connection of the external power supply is opposite between Fig.1 of this paper and the schematic figure in Ref.7. In Fig. 1, the positive electrode of the battery is connected to the thin film sample, whereas the negative electrode of the battery is connected in Ref.7. It is very confusing for readers. The author should clearly define the sign of the applied voltage in the experimental methods, rather than simply citing the literature.

Some minor issues:

(7) Page numbers is missing.

The page number of Ref. 8 is missing. The authors should add page numbers in reference.

(8) The titles of reference papers are missing.

The titles of Ref. 11 and Ref. 14 are missing. The authors should add the titles of papers in the reference.s

(9) The following two papers should be added to the reference as prior research on magnetoionics.

1. Nature 546, 124-128 (2017). (magnetoionic control using dual-ion; O^{2-} and H^{+})

2. Adv. Funct. Mater. 27, 1604990 (2017). (ON-OFF switching of ferromagnetism by Li-ion transport)

In the following, we provide detailed answers to each of the reviewers' comments (written in blue and italic font). The added and modified parts appear in the revised version of the manuscript over a yellow background:

Reviewer #1

In the manuscript "Voltage-driven motion of nitrogen ions: a new paradigm for magneto-ionics" the authors set out to demonstrate a new type of a magneto-ionic (magnetoelectric) system driven by Nitrogen ions rather than conventionally used (most studied) Oxygen. The authors argue and ultimately prove that voltage-controlled nitrogen mediated redox reactions - bringing about the magnetoelectric coupling - require lower threshold voltages and exhibit enhanced rates and cyclability.

To elucidate the phenomena behind the magnetization switching and phase transitions the authors have chosen a wide-ranging and well-thought set of characterization techniques. They inquired in detail into the structure, microstructure, elemental mapping, magnetic characteristics versus applied voltage etc. The conclusions reached are sound and sufficiently supported by the experimental data. The manuscript is clearly structured with the appropriate choice of references.

In particular, I find very appealing the combination of the semiconductor (or weakly metallic) material, undoubtedly resulting in the smooth and uniform redox process during the cycling, with the smart choice of the active ion.

In summary, this is a very interesting study; it should be published after addressing some remaining concerns pointed out below.

We do appreciate that Reviewer #1 considers our work very interesting and publishable after minor revision.

Some specific points recommended for considering:

*1) As the authors concluded the full chemical reduction of the CoN film shown in the report (i.e. conversion from CoN to Co metal; for example at -50 V) is irreversible; on the other hand reversibility is demonstrated for the small voltages and shorter charging times (Fig. 2). I think it would be beneficial to discuss shortly - or in other words - an estimate of the actual thickness of the CoN participating actively in the reversible cycling. For example, let's make a very simple (may be too simple but worth trying anyway) approximation and take the experimental change of magnetization of 1emu/cm^3 (as in Fig 2.) and compare it with the total Co(N) magnetization of 637emu/cm^3 after full reduction (Table 1). Now taking 85 nm of the CoN film thickness one can estimate roughly the thickness of the magnetically active layer - $(1/637) * 85\text{nm} = 0.13\text{nm}$. Does that mean that in the reversible cycling only a few top monolayers of the CoN are involved? How deep one can actually go with the CoN reduction to retain a full reversibility? Perhaps it would be instructive for the reader to shortly address those points since at the end the strength of the magnetic response to the electric stimulus (or strength of the magnetoelectric coupling) depends on those conditions.*

Reviewer #1 raises several interesting points regarding voltage cyclability in our system. His/her simple but representative approximation to estimate the magnetically active region indeed indicates that only a few monolayers are involved in the generation of a magnetization of 1emu/cm^3 . This actually corresponds to the magnetic moment induced close to the threshold voltage (where magneto-ionic effects are triggered) if this voltage is applied for a relatively short time (a few minutes). As supported by the HAADF-STEM/EELS and PALS characterization, N migration begins at

the upper part of the CoN layer, *i.e.*, at the boundary between the working electrode (the sample) and the electrolyte (which acts as N reservoir). With time, N ions from underneath regions become also activated. For example, Fig. 1g reveals that a magnetization of around 18 emu/cm³ can be generated in CoN if the threshold voltage is applied for about 1 hour. This corresponds to approximately 2.5 nm and the behavior is still reversible. In any case, reversibility is constrained mainly due to the limited solubility of nitrogen in propylene carbonate (see, *e.g.*, Chen, W. *et al.*, *Appl. Energy* **240**, 265–275 (2019)). An analogous reasoning can be made for oxygen from Co₃O₄ due to its limited solubility in propylene carbonate (see, *e.g.*, Read, J. *et al.*, *J. Electrochem. Soc.* **150**, A1351–A1356 (2003)). These low solubility limits cause N₂ (or O₂) bubbling to appear once propylene carbonate reaches nitrogen (or oxygen) supersaturation. The N₂ (or O₂) gases forming bubbles are released to the atmosphere and cannot be recovered. It might be that, with improved ion reservoirs (for instance, with solid state ionic conductors), reversibility could be enhanced in the future.

The following paragraphs and references have been added in the Results section (pages 5 and 9):

“For both systems, when gating at –50 V, bubbling occurs. Oxygen (in Co₃O₄) and nitrogen (in CoN) gas evolution is most likely the major source of bubbling. When a large negative voltage is applied, high concentrations of oxygen and nitrogen anions accumulate near the counter-electrode. If their concentration is high enough, bubbles form when the oxygen (or nitrogen) solubility limits^{36,37} are exceeded. When this happens, these elements are irreversibly lost from the system (sample + electrolyte) and this is the main reason causing the lack of reversibility of magneto-ionic effects when positive voltage is applied.”

“As a first approach, a simple calculation considering the overall magnetization induced in CoN when applying –50 V (*i.e.*, 637 emu/cm³, Table 1, when the whole CoN is affected, as will be shown in the forthcoming section) suggests that during this reversible cycling (1 emu/cm³, Fig. 2), only the uppermost 0.1-0.2 nm of the CoN layer are involved in the observed reversible effect. If the threshold negative voltage is applied for longer times (Fig. 1g), the induced magnetization results from nitrogen ion migration of an equivalent CoN thickness of 2-3 nm. Similar affected thicknesses during reversible voltage-driven magnetization cycling can be estimated for Co₃O₄.”

36. Read, J., Mutolo, K., Ervin, M., Behl, W., Wolfenstine, J., Driedger, A. & Foster, D. Oxygen transport properties of organic electrolytes and performance of lithium oxygen battery. *J. Electrochem. Soc.* **150**, A1351–A1356 (2003).

37. Chen, W., Chen, M., Yang, M., Zou, E., Li, H., Jia, C., Sun, C., Ma, Q., Chen G. & Qin, H. A new approach to the upgrading of the traditional propylene carbonate washing process with significantly higher CO₂ absorption capacity and selectivity. *Appl. Energy* **240**, 265–275 (2019).

2) In Fig. 1f,g the reaction threshold voltages are estimated and applied for 1 hour (chemical reduction). Could you please comment on the fact that in one hour Co₃O₄ reaches the magnetization of 200 emu/cm³ while CoN shows only 15 emu/cm³. How this could be reconciled with the generally faster magnetic response of the CoN over Co₃O₄ (Fig. 1d).

We would like to thank the reviewer for pointing out this issue. In fact, the determination of the exact threshold voltage is not so straightforward due to the interplay between voltage and time. Actually, after looking at Fig. 1f from the original version of the manuscript, it seemed that the magnetic signal was already increasing after applying –6 V. This observation and, particularly, the reviewer’s comment made us repeat the same experiment for Co₃O₄ but using –6 V instead of –8 V.

The result is shown in the new version of Fig. 1f and the corresponding hysteresis loops in the new version of Fig. S3.

Fig. 1 of the revised manuscript.

Supplementary Fig. S3 of the revised manuscript.

The new measurements reveal that after a better estimation of the threshold voltage for Co_3O_4 (-6 V instead of -8 V), both layers attain similar saturation magnetization values after the voltage is applied for sufficiently long times. Remarkably, comparison of the new Fig. 1f with Fig. 1g further corroborates that magneto-ionic rates are faster for CoN than for Co_3O_4 . We thank the reviewer for making us realize about this apparent contradiction.

3) A question out of curiosity: When all the CoN film is fully converted to Co metal, why is the measured Co magnetization after -50 V treatment (Table 1 M_s of 637) - less than half of the Co bulk value (bulk M_s of about 1400 emu/cm^3)? Perhaps the authors could shortly comment on it in the manuscript.

We also noticed this reduction of M_s . We believe that when nitrogen and oxygen diffuse towards the electrolyte, a large amount of vacancies and other structural defects are created in the films. In fact, the film thickness does not considerably reduce when negative voltage is applied, which means that the treated films (after applying negative voltage) have a lower density and possibly nanoporosity (due to coalescence of vacancies). Since it is not easy to quantify the nanoporosity degree or the amount of vacancies or regions with lower density, the estimated values of saturation magnetization are given after normalizing to the total volume of the film (*i.e.*, no correction is applied to account for porosity or the aforementioned structural defects). By doing so, the net volume of ferromagnetic phase is probably overestimated, and this results in magnetization values which are lower than for bulk, fully dense, cobalt.

A detailed analysis of the types of vacancies generated during electrolyte gating of Co_3O_4 films was reported in our previous publication (*ACS Nano* **12**, 10291–10300 (2018)). Complex vacancies, involving groups of various atoms, are formed during magneto-ionic experiments. The resulting Co is highly nanocrystalline or even amorphous (particularly the uppermost portions of the films in direct contact with the electrolyte). Indeed, a decrease of density (decrease of Co signal) is obvious in the TEM image of the CoN sublayer 1 in Fig. 3g, possibly due to the occurrence of free volume or nanoporosity. Note that a reduction of M_s has, in fact, been reported in amorphous Co nanoparticles compared to bulk fully dense Co (see for example *Langmuir* **25**, 10209–10217 (2009)).

To make this point clearer, we have now stated in the manuscript (Table 1, caption, page 8) that the M_s values are obtained by normalizing by the nominal film thickness, without taking into consideration eventual nanoporosity or vacancies generated during magneto-ionic treatments.

“Note that M_s has been obtained by normalizing the magnetic moment to the nominal film thickness, without taking into consideration the formation of vacancies or eventual nanoporosity.”

4) Why actually CoN is claimed to be semiconducting here (page 26) ? “Fig. S4) reveal their insulating and semiconductor character, respectively. This allows the layers to hold electric fields across them, which is a necessary condition for magneto-ionic phenomena.” Fig. S4 actually suggests a metallic like electronic transport down to 100 K. The CoN resistivity at room temperature is low – $4 \cdot 10^{-4}$ Ohm cm (for instance metallic alloys nichrome is of 10^{-4} ohm cm). If the CoN is metallic-like at room temperature that would mean a creation of the Helmholtz double layer at the CoN interface (no electric field penetration into the bulk of the CoN). Perhaps this would explain why - contrary to Co_3O_4 case - the redox reaction front moves into the material in a uniform way; this would mean a conversion reaction (CoN to Co) front moving with the Helmholtz double layer into the material interior; please address this issue and maybe consider discussing it in the manuscript.

The reviewer raises an interesting point concerning the electric properties of CoN. He/she is right that Fig. S4 suggests a metallic-like electronic transport down to 100 K (with positive dp/dT). Below this temperature, the sign of the dp/dT slope becomes negative, evidencing a complex electric transport behavior. We performed resistivity measurements on sputtered CoN layers grown at different N_2 pressures and the overall temperature dependence of resistivity is found to depend on the N content in the samples (see figure below). While only a negative slope is observed for the 75% N_2 film, a positive slope is measured in the whole temperature range for the film grown with 25% N_2 . In any case, the resistivity values at room temperature are higher than the values in metals such as Cu, Fe or Pt (which are of the order of 10^{-6} - 10^{-5} Ω cm) and lower than values obtained in wide bandgap semiconductors. Nonetheless, Co-rich CoN films are often claimed to be metallic in the literature (e.g., *Inorg. Chem. Front.* **3**, 236–242 (2016); *Chem. Mater.* **30**, 5941–5950 (2018)).

To take this complex behavior into account, we have now replaced “semiconducting behavior” by “semiconducting/metallic behavior” throughout the manuscript when referring to the electric properties of CoN.

Importantly, as pointed out by the reviewer, the electric properties of the investigated materials are likely to play a significant role on the observed dissimilar magneto-ionic phenomena between the two investigated films. While electric field in metals is effectively screened at their outmost surface (electrostatic shielding), Co oxides are good insulators, and we believe that ionic movement in Co_3O_4 proceeds through dielectric breakdown (local ionization), due to the strong covalent character of the bonding. In turn, CoN has orders of magnitude smaller resistivities compared to Co_3O_4 , and the number of factors to consider (electric field screening, chemical bonding) are larger. Nonetheless, the metallic-like character of CoN at room temperature might indeed explain why, contrary to the Co_3O_4 case, the redox reaction front in this case moves into the material in a uniform way (plane-wave-like migration front). We believe a clear description of these effects requires a more detailed investigation, which we will certainly undertake in the future. To address the reviewer’s observation, the following sentence has been added in the Discussion of the new version of the manuscript (page 18):

“The dissimilar electric properties of CoN and Co_3O_4 are also likely to play a role in the way ions diffuse in the two layers.”

Reviewer #2

This paper reports electrical control of magnetism by voltage-driven ion transport (magneto-ionic control) for two-types of cobalt compounds; CoN and Co₃O₄. The authors control transport of nitrogen ion or oxygen ion by electrolyte-gating method in a CoN- and Co₃O₄-films, respectively. They have found reversible voltage-driven magnetic switching behavior between paramagnetic and ferromagnetic states at room temperature. By comparing the responsiveness of magnetization to gating voltage for CoN with that of Co₃O₄, they have also found lower operating voltage and the enhanced cyclability in CoN. The superior electrical magnetism-controllability of CoN is clearly demonstrated, and the results would be promising for the future application of magneto-ionics in devices. However, there are several comments and questionable points, which are shown below, on this paper in its present form. In particular, the electrochemical assessment of the electrolytic cell used for the measurements seems to be insufficient. In my opinion, they should be made clear before the manuscript is suitable for publication.

We thank the reviewer for his/her positive feedback, highlighting the superior responsiveness of CoN compared to Co₃O₄ and considering that the results of our work are promising for future studies and applications of magneto-ionics.

(1) Redox potential measurements of the electrolytic cell should be conducted.

In this work, the authors have applied a voltage as high as -50V to the cell. Such high voltage should induce a variety of electrochemical reaction not only in magneto-ionic target (CoN and Co₃O₄ film) but also in the components of the used electrolytic cell; redox reaction of liquid electrolyte and Cu/Ti-metal film. At least, the redox potentials of CoN and Co₃O₄ should be checked by measuring cyclic voltammetry. They should directly correlate with the responsiveness of magnetization to the applied voltages.

Following the reviewer's suggestion, the components of the electrochemical cell have been subjected to a series of cycling voltammetry experiments using propylene carbonate in order to test the existence of eventual electrochemical reactions. First, the potential window of the propylene carbonate electrolyte was confirmed using a Pt working electrode in the largest possible voltage window allowed by our instrumentation: -40/40 V (please note that to record the cyclic voltammetry curves a power supply different from the one used to run magneto-electric measurements was employed). As it can be seen from the figure below (Figure S6 in the revised version of manuscript), the lack of oxidation/reduction peaks and the small currents involved (with currents densities of the order of $\mu\text{A}/\text{cm}^2$) indicate the absence of charge transfer reactions at the surface of the Pt working electrode. Importantly, no supporting electrolyte was used either here or during our magneto-ionic experiments. Therefore, the analyzed electrolyte is a priori resistive to charge transfer, contrary to aqueous media, Li-containing electrolytes or ionic liquids widely used in other magnetoelectric studies (e.g. *Small* **15**, 1904523 (2019); *Nano Lett.* **16**, 583-587 (2016); or *Adv. Mater.* **30**, 1703908 (2018); etc.). It should be noted, nevertheless, that we have sometimes encountered polymerization effects in the working electrode when using propylene carbonate at even larger voltages (e.g., +100 V).

Fig. S6 | Cyclic voltammograms of Pt working electrode in anhydrous propylene carbonate with traces of Na^+ and OH^- ions. Cyclic voltammograms using different potential windows are shown. The curves were recorded at 250 mV/s with a Pt wire pseudo-reference electrode. The arrows indicate the direction in which potential was scanned.

The reviewer is right in pointing out that any alteration of the Cu/Ti conducting layer underneath the targeted magneto-ionic materials, either Co_3O_4 or CoN , could have some effects on the performance of the magneto-ionic systems. As demonstrated in a previous work from our group (see de Rojas, J. *et al. Adv. Funct. Mater.* **30**, 2003704 (2020)), the electrical conductivity of the buffer layer can speed up/slow down the rate of induced magnetization as well as the total generated magnetization during the magneto-ionic processes. For this reason, the stability of the Si/Ti/Cu substrate immersed in the propylene carbonate electrolyte has been also tested. The results, shown in the Figure below (left), demonstrate the absence of distinct Cu oxidation/reduction peaks and very low current densities (background current) varying smoothly with potential, similarly to what was observed when cycling propylene carbonate using the Pt electrode. To further corroborate this point, the Si/Ti/Cu substrate was then immersed in aqueous 1M NaOH solution and a cyclic voltammogram hysteresis loop was recorded under the same testing conditions. In this case, as it can be seen from the right panel of the figure below, several well-defined peaks corresponding to the formation of Cu oxides and hydroxides are observed. In addition, the current densities in the aqueous electrolyte are of the order of mA (much larger than when using propylene carbonate, where currents are of order of μA). This suggests that faradaic processes dominate over capacitive ones in an aqueous electrolyte, while there is no evidence of charge transfer reactions occurring in Si/Ti/Cu immersed in propylene carbonate (anhydrous) electrolyte. We should also point out that during the magneto-electric measurements there is no Cu exposed to the electrolyte. Thus, there is no means the Cu substrate can get oxidized or reduced with the propylene carbonate.

Cyclic voltammograms of Cu in anhydrous propylene carbonate with traces of Na^+ and OH^- ions (left) and in aqueous 1 M NaOH solution (right). The arrows indicate the direction in which potential was scanned.

Additionally, the electrochemical stability of the underneath Cu/Ti films was evaluated by TEM. As can be observed from electron energy loss spectroscopy images (EELS) in Fig. 3d and 3h of the manuscript, which belong to the samples treated during 75 min to -50 V, no oxygen or nitrogen signal is detected from the substrate Cu layer after the electrolyte gating process. This confirms the stability of the substrate layers and, therefore, corroborates that the induced effects are due to ion migration in the tested magneto-ionic materials.

Finally, cyclic voltammetry tests were performed on the CoN and Co_3O_4 samples immersed in anhydrous propylene carbonate. The potential window was set from -20 to 20 V in order to detect any oxidation/reduction reactions taking place after overcoming the threshold magneto-ionic voltage (-4 V and -6 V, as evidenced from the magneto-electric measurements, for CoN and Co_3O_4 , respectively). To acquire each voltammetry curve, the voltage was kept constant for 10 min in each point before sweeping voltage (due to a very slow electrode kinetics), mimicking the conditions of the magneto-ionic experiments. Unfortunately, as can be seen in the figure below (Figure S10 of the revised version of manuscript), it is not possible to strictly determine the redox potentials of CoN and Co_3O_4 , or to correlate the threshold voltage for magneto-ionics with electrochemical potentials for the oxidation/reduction of Co. The absence of clear maxima and the low values of current density indicate the occurrence of non-faradaic phenomena revealing that, despite the oxygen/nitrogen ion migration, these processes constitute a non-conventional redox reaction. Previous works from our group proved that the currents involved in the magneto-ionic events gated with anhydrous electrolytes are of the same order of magnitude (Navarro, C. *et al.*, *ACS Appl. Mater. Interfaces* **10**, 44897–44905 (2018)). In any case, the cyclic voltammetry results are consistent with the observed magneto-ionic effects since the non-linear increase in the negative current upon application of negative voltage could be attributed to the generation of metallic Co. However, in our system, the changes in the oxidation state of Co are not caused by the standard redox reactions but are

magneto-ionically driven with oxygen/nitrogen ion exchange between the cobalt oxide/nitride and the non-aqueous electrolyte (as evidenced by the detailed structural characterization provided).

To address these points the following sentences have been included in the revised version of the manuscript:

“Partial decomposition of electrolyte gate (*i.e.*, propylene carbonate) at high voltages cannot be completely ruled out³⁸, although no electron-transfer reactions can be observed in the cyclic voltammetry curve when using Pt as working electrode (Fig. S6) and no bubbling was observed in this case.” (page 5)

“Electric-field induced oxygen ions exchange with the liquid electrolyte is the main reason of the increased current densities observed in the cycling voltammetry curves (Fig. S10).”

“Similar to Co_3O_4 , these processes are essentially non-Faradaic and do not cause pronounced peaks in the cyclic voltammetry curves (Fig. S10).” (page 11)

Fig. S10 Cyclic voltammetry curves in anhydrous propylene carbonate with Na^+ and OH^- ions for CoN and Co_3O_4 films. The arrows indicate the direction in which potential was scanned.

The following reference has been added when introducing magneto-ionics (page 3):

17. Molinari, A., Hahn, H. & Kruk, R. Voltage-controlled on/off switching of ferromagnetism in manganite supercapacitors. *Adv. Mater.* **30**, 1703908 (2018).

(2) Does the Cu-layer not react with migrated oxygen or nitrogen ions?

The working electrode of electrolytic cell consists of Cu-metal. I am afraid that the Cu-layer might be oxidized under the high voltage and the oxidized Cu might react with the migrated oxygen or nitrogen ions. In such case, the formation of copper oxides and nitrides would affect the responsiveness of magnetization. Did the authors confirm the stability of the copper layer during voltage sweeping

process? In case that oxidation of Cu occurs, the oxidation current would be detected during voltage sweeping process.

In the electrolytic cell configuration utilized in our experiments (*i.e.*, Cu underlayer and Pt wire as working and counter electrodes, respectively), magneto-ionics is enabled by inducing oxygen and nitrogen ion migration towards the electrolyte by applying negative voltages. Specifically, ion migration starts at the top of the Co_3O_4 and CoN films (*i.e.*, at the boundary with the electrolyte, which acts as ion reservoir) and, with time, ions from underneath regions become activated. This way the Cu buffer layer cannot be oxidized or nitrided, since the ions move towards the electrolyte, not towards Cu. As observed in the electron energy loss spectroscopy images of Figs. 3d and 3h, which correspond to the films Co_3O_4 and CoN treated at -50 V for 75 min, respectively, neither oxygen nor nitrogen signals are detected within the Cu layer, confirming its chemical stability.

Upon application of positive voltages, used to invert the magneto-ionic process (*i.e.*, always applied after negative voltages), the ions oxygen or nitrogen ions previously dissolved in the electrolyte are reincorporated in the films, oxidizing/nitriding the generated Co-rich regions back to the non-magnetic oxide or nitride phases. However, for the conditions used in our experiments, we believe the ions from the solution cannot reach the Cu underlayer. Actually, if that would be the case, metallic Co may form again once Cu would integrate the oxygen or nitrogen from the Co_3O_4 or CoN layers. This would give rise to an increase of magnetization, which is not observed for positive voltages. In addition, as it was previously demonstrated with cycling voltammetry tests discussed in response to comment # 1, we do not observe currents high enough to be unambiguously attributed to the oxidation/reduction of the Cu substrate in anhydrous propylene carbonate.

Cu oxidation or nitriding may not be ruled out if a sufficiently high positive voltage was applied first. In that case oxygen or nitrogen ions would indeed tend to diffuse towards the Cu seed layer instead of the electrolyte. The reviewer is right in pointing out that any alteration of the Cu working electrode would definitely affect the induced magneto-ionic effects. A decrease in ionic motion and generated magnetic moment would be expectable if conductivity of the Cu layer was reduced.

(3) How did the authors identify the gases produced under the gating voltage of -50 V as O_2 and N_2 ?

In Page 5, the occurrence of bubbling under the gating voltage of -50 V is reported for both Co_3O_4 and CoN. The authors identify the generating gases as O_2 and N_2 from Co_3O_4 and CoN, respectively. However, in such voltage as high as -50 V, solvent of electrolyte (propylene carbonate) might be electrochemically decomposed to generate gases. Was the electrochemical window of electrolyte confirmed? The author should state how they distinguish the evolution of O_2/N_2 gas from that of other possible origins.

The stability of the electrolyte was assessed by cycling voltammetry measurements, which indicated the occurrence of capacitive processes in the potential window $-40/40$ V (see response to comment #1). Thus, in this potential range, the formed bubbles are highly likely to be linked to the formation of oxygen and nitrogen molecule gases, since there is no evidence of electron charge transfer reactions. We agree with the reviewer that a partial cathodic decomposition of propylene carbonate may occur at high voltages, being propene (see Eichinger, G. J. *Electroanal. Chem. Interf. Electrochem.* **74**, 183–193 (1976)) a source of gas (bubbling). However, no bubbling was observed when performing analogous magneto-ionic experiments on metallic alloy systems or during cyclic voltammetry using Pt as working electrode. Moreover, bubbling was, in fact, more prominent for CoN (compared to

Co₃O₄), where magneto-ionic effects are in fact more pronounced. Note that for negative voltages, oxygen (or nitrogen) ions are released to the electrolyte, get dissolved, and eventually form bubbles at the counter-electrode once the solubility limits for O₂ and N₂ are exceeded (Refs. 35 and 36 of the revised manuscript). Due to the rather limited solubility of both gases in propylene carbonate, it is indeed expected that for very long gating voltages and time, the ions are released in the form of gas from the electrolyte into the atmosphere.

The following sentences have been added in the revised version of the manuscript (page 5):

“For both systems, when gating at –50 V, bubbling occurs. Oxygen (in Co₃O₄) and nitrogen (in CoN) gas evolution is most likely the major source of bubbling. When a large negative voltage is applied, high concentrations of oxygen and nitrogen anions accumulate near the counter-electrode. If their concentration is high enough, bubbles form when the oxygen (or nitrogen) solubility limits^{36,37} are exceeded. When this happens, these elements are irreversibly lost from the system (sample + electrolyte) and this is the main reason causing the lack of reversibility of magneto-ionic effects when positive voltage is applied. Partial decomposition of electrolyte gate (*i.e.*, propylene carbonate) at high voltages cannot be completely ruled out³⁸, although no electron-transfer reactions can be observed in the cyclic voltammetry curve when using Pt as working electrode (Fig. S6) and no bubbling was observed in this case. Bubbling is, in fact, more pronounced for CoN than for Co₃O₄, where the magneto-ionic response is stronger.”

36. Read, J., Mutolo, K., Ervin, M., Behl, W., Wolfenstine, J., Driedger, A. & Foster, D. Oxygen Transport Properties of Organic Electrolytes and Performance of Lithium Oxygen Battery. *J. Electrochem. Soc.* **150**, A1351–A1356 (2003).

37. Chen, W., Chen, M., Yang, M., Zou, E., Li, H., Jia, C., Sun, C., Ma, Q., Chen G. & Qin, H. A new approach to the upgrading of the traditional propylene carbonate washing process with significantly higher CO₂ absorption capacity and selectivity. *Appl. Energy* **240**, 265–275 (2019).

38. Eichinger, G. Cathodic decomposition reactions of propylene carbonate. *J. Electroanal. Chem. Interf. Electrochem.* **74**, 183–193 (1976).

(4) Are the Na⁺ ions in liquid electrolyte inserted into the CoN/Co₃O₄ film or not during voltage sweeping process?

The liquid electrolyte used in this work contains Na⁺-ions. In the recent years, the Na⁺-ion is intensively investigated as charge carrier species in ion-batteries. In the present case, can the authors exclude the possibility Na⁺-ion insertion into CoN/Co₃O₄? In the case that the Na⁺-ion insertion occurs, Na⁺-ion insertion would contribute to redox reaction in CoN and Co₃O₄, and thus it would affect the responsiveness of magnetization to the applied voltages.

We thank the reviewer for highlighting this important point. It is well known that Na⁺, like Li⁺, can mediate in redox reactions and therefore, modify the ferromagnetic properties of target materials (see Dasgupta, S. *et al. Adv. Mater.* **26**, 4639–4644 (2014); Dwibedi, D. *et al. AIP Conf. Proceed.* **1832**, 130041 (2017)). The concentration of Li⁺ in the electrolyte used in those experiments is in the molar range (see Wei, G. *et al. RSC Adv.* **7**, 2644–2649 (2017)). However, in our electrolyte, Na⁺ is only present in much lower concentrations (*i.e.*, ppm quantities, see Quintana, A. *et al. Adv. Funct. Mater.*

27, 1701904 (2017)) and its main function is to preserve the anhydrous character of the solvent (see Weisheit, M. *et al. Science* **315**, 349-351 (2017)). Further evidence of Na^+ not taking place as a mechanism in Co_3O_4 reduction into Co, was demonstrated in a previous publication from our group (see Fig. S12 in Quintana, A. *et al. ACS Nano* **12**, 10291–10300 (2018)). In that work, magneto-ionic experiments were performed in similar Co_3O_4 films. In order to discard Na^+ mediated phenomena, we performed EELS analysis on the Na K edge, which revealed no measurable signal (*i.e.*, no Na^+ inclusion or its presence was well below the detection limit), thereby essentially discarding any Na^+ reduction mediation process.

(5) In the Ab initio calculations, were the crystal structures of Co_3O_4 and CoN reproduced in the lattice relaxation process after oxygen or nitrogen insertion?

For comparison of the formation energy of Co-O and Co-N, the calculation for the oxygen and nitrogen insertion to the hexagonal closed-packed Co slab has been conducted (In Page 19 (Methods, Ab initio calculations). However, the sublattice-structures of Co in Co_3O_4 and CoN are different from the hexagonal closed-packed structure. In the relaxation process of the atomic coordinates, were the crystal structures of Co_3O_4 and CoN reproduced? If not, is it plausible to use the calculated values of energy barriers for explanation of the observed superior electrical magnetism-controllability of CoN?

We thank the reviewer for this insightful comment and questions. The primary purpose of our *ab initio* calculations was to provide insights supporting the experimental observation and compare the nitrogen versus oxygen magneto-ionics. For that, we employed a simplified approach by calculating straightforwardly the energy barrier that an O or N atom needs to overcome to get inserted inside a Co film. Due to the limitations of the applied density-functional theory and, in particular, the nudged elastic band method, the CoN and Co_3O_4 crystallographic structures were not reproduced, thus starting initially from the Co film while calculating the presented energetics. We agree with the reviewer that reproducing those crystals would be the ideal approach; however, with the chosen method we were able to provide both a good explanation of the ionic energetics and reasonable agreement with the experiment. More importantly, the calculation allowed estimating the critical electric field needed to overcome the barrier. It is worth to note that the same method was efficiently used to describe the voltage control of the magnetic anisotropy by O migration at Fe/MgO [Ibrahim, F. *et al., Phys. Rev. B* **98**, 214441 (2018)]. Therefore, we believe that the applied approach is plausible to compare the O and N magneto-ionics.

We also agree with the reviewer that the Co sublattices in CoN and Co_3O_4 are different from Co HCP structure. Due to the observation of both HCP (0001) and FCC (111) Co phases in the experimental samples [supplementary Table S3] and following the reviewer's comment, we performed additional calculations using a Co FCC (111) film. The obtained results are presented in the figure below and show larger energy barriers in the case of FCC-Co compared to its HCP-Co counterpart, but our main conclusions on a lower energy barrier in case of N (compared to O) still holds, independently of the Co phase. Indeed, taking into account an energy barrier $\Delta V = 1.37$ eV/atom (1.85 eV/atom) for FCC Co-N (Co-O), respectively, the corresponding critical electric field needed to overcome this barrier is estimated as $E_c = 6.3$ V/nm (8.5 V/nm) that is still in reasonable agreement with experiment.

Following the reviewer's comments, we have updated Fig. 5(b) and the corresponding discussion in the main text (pages 15, 16, 17):

"In turn, the calculated energy barriers between the two minima are 1.54 (1.85) and 1.14 (1.37) eV/atom for oxygen and nitrogen displacement into HCP (FCC) Co surface, respectively."

" E_c is found to be 8.1 (8.5) V nm⁻¹ and 5.3 (6.3) V nm⁻¹ for oxygen and nitrogen migration into HCP (FCC) Co, respectively, in good agreement with the onset voltages from magnetoelectric measurements (Figs. 1f and 1g) (thickness of the electric double layer < 1 nm). It is important to point out that due to the limitations of the applied density-functional theory and in particular the nudged elastic band method, the CoN and Co₃O₄ crystallographic structures were not reproduced, starting initially from the Co film while calculating the presented energy considerations. However, with the chosen method we were able to provide both a good explanation of the ionic energetics and reasonable agreement with the experiment. More importantly, the calculations allowed estimating the critical electric field needed to overcome the barrier. It is worth noting that the same method was efficiently used to describe the voltage control of the magnetic anisotropy by O migration at Fe/MgO²². Therefore, we believe that the applied approach is plausible to compare O vs. N magneto-ionics."

Fig. 5 | *Ab initio* calculations: magnetism in the Co-N system and Co-O vs. Co-N formation energy. Calculated total energy per atom, normalized to the minimum energy value, as a function of the displacement between the reference Co outermost surface atom and the inserted O or N atom. **Both HCP (0001) and FCC (111) surfaces are considered represented by closed and open symbols, respectively.** The black squares (red circles) correspond to the O (N) energetic path, respectively. The five-monolayer-thick Co slab is shown in the right panel, where the dashed line indicates the reference z position, which is the outermost Co surface monolayer.

(6) *The definition of the sign of the applied voltage should be clearly stated.*

In page 17 (Methods, magnetoelectric characterization), it is stated that the voltage was applied in the same fashion to that presented in references 7, 8 and 27. However, the direction of the battery

connection of the external power supply is opposite between Fig.1 of this paper and the schematic figure in Ref. 7. In Fig. 1, the positive electrode of the battery is connected to the thin film sample, whereas the negative electrode of the battery is connected in Ref.7. It is very confusing for readers. The author should clearly define the sign of the applied voltage in the experimental methods, rather than simply citing the literature.

We do agree with the reviewer. To avoid misunderstandings, citation to Refs. 7, 8 and 27 has been eliminated and the following sentence has been added in the Methods section (page 19):

“The sign of voltage was such that negative charges accumulate at the working electrode when negative voltage was applied (and vice versa for positive voltages). in a similar fashion to that presented in references 7, 8 and 27.”

Some minor issues:

(7) Page numbers is missing.

The page number of Ref. 8 is missing. The authors should add page numbers in reference.

Reference 8 has been now updated (note that only the “early view” online version was available at the time of the original submission of our manuscript):

8. de Rojas, J., *et al.* Boosting room-temperature magneto-ionics in a non-magnetic oxide semiconductor. *Adv. Funct. Mater.* **30**, 2003704 (2020).

(8) The titles of reference papers are missing.

The titles of Ref. 11 and Ref. 14 are missing. The authors should add the titles of papers in the reference.s

We thank the reviewer for this observation. The titles of these papers have been now added.

11. Dasgupta, S. *et al.* Intercalation-driven reversible control of magnetism in bulk ferromagnets. *Adv. Mater.* **26**, 4639–4644 (2014).

14. Vasala, S. *et al.* Reversible tuning of magnetization in a ferromagnetic Ruddlesden–Popper-type manganite by electrochemical fluoride-ion intercalation. *Adv. Electron. Mater.* **6**, 1900974 (2020).

(9) The following two papers should be added to the reference as prior research on magnetoionics.

1. *Nature* **546**, 124–128 (2017). (magnetoionic control using dual-ion; O²⁻ and H⁺)

2. *Adv. Funct. Mater.* **27**, 1604990 (2017). (ON-OFF switching of ferromagnetism by Li-ion transport)

We do agree with the reviewer. Thus, we have added these two publications in the reference list of the revised manuscript:

15. Lu, N. *et al.* Electric-field control of tri-state phase transformation with a selective dual-ion switch. *Nature* **546**, 124–128 (2017).

16. Taniguchi, K., Narushima, K., Sagawama, H., Kosaka, W., Shito, N., Miyasaka, H. In situ reversible ionic control for nonvolatile magnetic phases in a donor/acceptor metal-organic framework. *Adv. Funct. Mater.* **27**, 1604990 (2017).

Note that all references and figures in the Supplementary Information have been subsequently renumbered.

Reviewers' Comments:

Reviewer #1:

Remarks to the Author:

The authors have addressed in a satisfactory form the remarks and concerns of my review. No further changes are needed.

Reviewer #2:

Remarks to the Author:

Following my comments, the authors have carefully conducted the electrochemical assessment of the electrolytic cells. Their explanations based on the experimental results added in the revised manuscript are plausible, and I agree with their arguments. Concerning about the ab initio calculation, they have also done additional calculations for the oxygen and nitrogen migration into FCC Co lattice. I think the experimentally observed trend of higher mobility of nitrogen ions compared to oxygen ions is well supported qualitatively by the calculation.

The manuscript has been improved enough in the reviewing process, and I recommend publication this work in Nature Communications.

Some minor issues:

(1) What is the cause of the bending in the cyclic voltammetry curve in Figure S6?

In the cyclic voltammetry curve of Fig. S6, the cyclic voltammetry curve bends around +5 V and -5V. The author should comment about the possible origin for this point.

(2) Please check the onset voltage of Co₃O₄ in Table 1.

In Table 1, the onset voltage of Co₃O₄ is shown as -8V. However, the onset value in the main text is -6V and it might be an error in writing. So, please check the value.

(3) Please check the caption of Fig. S3.

In Fig. S3, the caption is written as follows,

"consecutive hysteresis loops under -8 V and -4 V gating and the corresponding recovery loops recorded at +8V and +4 V"

However, the voltage value displayed in the inset of Fig. S3a for Co₃O₄ is -6V and it might be typo. So, please check.

In the following, we provide detailed answers to each of the reviewers' comments (written in blue and italic font). The added and modified parts appear in the second revised version of the manuscript over a green background:

Reviewer #1

The authors have addressed in a satisfactory form the remarks and concerns of my review. No further changes are needed.

We do appreciate that Reviewer #1 considers our work ready for publication.

Reviewer #2

Following my comments, the authors have carefully conducted the electrochemical assessment of the electrolytic cells. Their explanations based on the experimental results added in the revised manuscript are plausible, and I agree with their arguments. Concerning about the ab initio calculation, they have also done additional calculations for the oxygen and nitrogen migration into FCC Co lattice. I think the experimentally observed trend of higher mobility of nitrogen ions compared to oxygen ions is well supported qualitatively by the calculation.

The manuscript has been improved enough in the reviewing process, and I recommend publication this work in Nature Communications.

We do appreciate that Reviewer #2 considers our work for publication upon addressing the comments below:

(1) What is the cause of the bending in the cyclic voltammetry curve in Figure S6?

In the cyclic voltammetry curve of Fig. S6, the cyclic voltammetry curve bends around +5 V and -5V. The author should comment about the possible origin for this point.

This bending can be possibly ascribed to absorption/desorption processes, mediated by surface impurities on Pt, slightly increasing the current. This could be minimized by an anodic pretreatment of Pt in acidic solution. Note that we only did some mechanical polishing of Pt, followed by ultrasonic cleaning in acetone, ethanol and milli-Q water. Nonetheless, given the low currents involved, this effect is difficult to be investigated in detail. However, the lack of clear peaks suggests that we can essentially rule out any electron transfer reaction as the origin of this bending.

(2) Please check the onset voltage of Co₃O₄ in Table 1.

In Table 1, the onset voltage of Co₃O₄ is shown as -8V. However, the onset value in the main text is -6V and it might be an error in writing. So, please check the value.

The onset voltage of Co₃O₄ is -6 V and it has been corrected in Table 1.

(3) Please check the caption of Fig. S3.

In Fig. S3, the caption is written as follows,

“consecutive hysteresis loops under -8 V and -4 V gating and the corresponding recovery loops recorded at +8V and +4 V”

However, the voltage value displayed in the inset of Fig. S3a for Co₃O₄ is -6V and it might be typo. So, please check.

The measurements on Co₃O₄ were performed at either -6 V or +6 V. The caption of Fig. S3 has been corrected.